# Two Novel *Lasiodiplodia* Species from Blighted Stems of *Acer truncatum* and *Cotinus coggygria* in China

**DOI:** 10.3390/biology11101459

**Published:** 2022-10-05

**Authors:** Guanghang Qiao, Juan Zhao, Juanjuan Liu, Xiaoqian Tan, Wentao Qin

**Affiliations:** 1Institute of Plant Protection, Beijing Academy of Agriculture and Forestry Sciences, Beijing 100097, China; qghang98@126.com (G.Q.); zhaojuan119882@163.com (J.Z.); juanjuan_9215@163.com (J.L.); tanxiaoqian159@163.com (X.T.); 2College of Forestry, Beijing Forestry University, Beijing 100083, China; 3College of Life Sciences, Yangtze University, Jingzhou 434025, China

**Keywords:** Botryosphaeriaceae, morphology, phylogeny, taxonomy

## Abstract

**Simple Summary:**

*Lasiodiplodia* species are plurivorous plant pathogens found worldwide, especially in tropical and subtropical regions, that result in fruit and root rot, die-back of branches and stem canker, etc. During the exploration of the fungal diversity of blighted stem samples collected in northern China, two new *Lasiodiplodia* species, *L. acerina* G.H. Qiao & W.T. Qin and *L. cotini* G.H. Qiao & W.T. Qin, were discovered based on integrated studies of phenotypic features, culture characteristics and molecular analyses. They were described and illustrated in detail. This work provided a better understanding of the biodiversity, phylogeny and established concepts of the genus *Lasiodiplodia*.

**Abstract:**

The *Lasiodiplodia* are major pathogens or endophytes living on a wide range of plant hosts in tropical and subtropical regions, which can cause stem canker, shoot blight, and rotting of fruits and roots. During an exploration of the stem diseases on *Acer truncatum* and *Cotinus coggygria* in northern China, two novel species of *Lasiodiplodia*, *L. acerina* G.H. Qiao & W.T. Qin and *L. cotini* G.H. Qiao & W.T. Qin, were discovered based on integrated studies of the morphological characteristics and phylogenetic analyses of the internal transcribed spacer region (ITS), translation elongation factor 1-α (*TEF1-α*), beta-tubulin (*TUB2*) and RNA polymerase II subunit b genes (*RPB2*). *Lasiodiplodia acerina* is a sister taxon of *L. henannica* and distinguishable by smaller paraphysis and larger conidiomata. *Lasiodiplodia cotini* is closely related to *L. citricola* but differs in the sequence data and the size of paraphyses. Distinctions between the two novel species and their close relatives were compared and discussed in details. This study updates the knowledge of species diversity of the genus *Lasiodiplodia*. Furthermore, this is the first report of *Lasiodiplodia* associated with blighted stems of *A. truncatum* and *C. coggygria* in China.

## 1. Introduction

*Lasiodiplodia*, established in 1896, is a member of the family Botryosphaeriaceae [1]. Species in the genus *Lasiodiplodia* have been associated with different plant diseases including fruit and root rots, die-back of branches and stem cankers. The type species of *Lasiodiplodia* (*L. theobromae*) was regarded as one of the cosmopolitan, plurivorous pathogens mainly inhabiting tropical and subtropical regions [2,3].

The main morphological characteristics of *Lasiodiplodia* include hyaline, smooth, cylindrical to conical conidioenous cells, which produce subovoid to ellipsoid-ovoid conidia and the conidia are hyaline without septa or dark-brown with single septae [4]. Species in the genus *Lasiodiplodia* were mostly differentiated based on the characteristics of the conidia and paraphyses [5]. Some other morphological characteristics, such as annelations of conidiogenous cells, the dimensions and papillate nature of conidiomata, septate and pigmented conidia as well as the pycnidial paraphyses have been gradually used to recognize the *Lasiodiplodia* species, but to what extent these characteristics are phylogenetically significant warrants further investigation [6].

The Genealogical Concordance Phylogenetic Species Recognition (GCPSR) concept is widely used to delineate different fungal species. This approach relies on determining the concordance between multiple gene genealogies and delimiting species where the branches of multiple trees display congruence [7]. The widespread application of phylogenies based on ITS, *TEF1-α*, *TUB2* and *RPB2* genes promotes the accurate identification of species in the genus *Lasiodiplodia*, and more and more species have been successively introduced over the years; at present, more than 70 *Lasiodiplodia* species have been identified [8,9,10]. Among them, some species have been introduced almost entirely on the basis of DNA sequence phylogenies. Although the phylogenies were derived from the analysis of multiple loci, some species were introduced only on the basis of minor differences in only one locus, and some species cannot be clearly separated phylogenetically [11,12,13]. Several accepted *Lasiodiplodia* species (*L. brasiliense*, *L. laeliocattleyae*, *L. missouriana*, *L. viticola*) may be hybrids based on a detailed phylogenetic analyses of five loci from 19 *Lasiodiplodia* species [14].

To provide a better understanding of *Lasiodiplodia* species diversity in China, recent collections of the genus on *Acer truncatum* and *Cotinus coggygria* were examined. Two previously unrecognized *Lasiodiplodia* species were discovered based on integrated studies of phenotypic features, culture characteristics and phylogenetic analyses of the combined sequences of ITS, *TEF1-α*, *TUB2* and *RPB2*. Detailed comparisons were made between the new taxa and their close relatives.

## 2. Materials and Methods

### 2.1. Isolates and Specimens

Cultures were isolated from the blighted stems of *Cotinus coggygria* and *Acer Truncatum* collected from Beijing, China, from 2018 to 2019. Stem segments (0.5 cm × 0.5 cm × 0.2 cm) were cut from the boundary of the lesion or dead tissues, surface sterilized subsequently and incubated on potato dextrose agar (PDA, peeled potatoes 200 g, glucose 20 g, agar 18 g, add water to 1 L) at 25 °C for fungal isolation [15]. Specimens, purified cultures and the ex-type strains were deposited in the culture collection of Institute of Plant Protection, Beijing Academy of Agriculture and Forestry Sciences.

### 2.2. Morphology and Growth Characterization

Morphological characterization of colonies, such as colony appearance, color and spore production were observed and recorded following the method of previous studies [5,11,16] on three media (PDA, malt extract agar (MEA, malt extract 20 g, agar 18 g, add water to 1 L) and synthetic nutrient-poor agar (SNA, monopotassium phosphate 1 g, potassium nitrate 1 g, Magnesium sulfate heptahydrate 0.5 g, potassium chloride 0.5 g, glucose 0.2 g, saccharose 0.2 g, agar 20 g, add water to 1 L)) with each isolate three replicates. Microscopic characteristics were recorded based on 20 paraphyses, 20 conidiogenous cells and 50 conidia on PDA at 25 °C in darkness. Photographs were taken from material mounted in lactic acid with Axiocam 506 color microscope (Carl Zeiss, Aalen, Germany) using Zeiss Imager Z2 software. The new species were established based on the guidelines outlined by Jeewon and Hyde [17].

### 2.3. DNA Extraction, PCR Amplification and Sequencing

Purified cultures were incubated on PDA with cellophane for 5 days at 25 °C in darkness. Genomic DNA was extracted using the TsingKe Plant Genomic DNA Extraction Kit^®^ following the manufacturer’s protocol (Beijing, China). The ITS, *TEF1-α*, *TUB2* and *RPB2* gene sequences were amplified and sequenced using primer pairs ITS1/4 [18], EF1-728F/986R [19], Bt2a/2b [20] and RPB2-LasF/R [14], respectively. Each PCR reaction (25 μL) consisted of 1 μL 5–10 ng DNA, 22 μL TsingKe Golden Star T6 Super PCR Mix (1.1×) and 1 μL of each primer. PCR amplification followed the manufacturer’s protocol of TsingKe Golden Star T6 Super PCR Mix (Beijing, China), and products were sequenced by Beijing TsingKe Biotech Co. Ltd. (Beijing, China).

### 2.4. Sequence Alignment and Phylogenetic Analyses

Sequences of the investigated *Lasiodiplodia* species excluding those of our two new species for phylogenetic analyses were obtained from the NCBI using Tbtools v. 1.09876 [21] (Table 1). Sequences were assembled, aligned and manually adjusted with BioEdit v.7.2.5 [22]. To identify the phylogenetic positions of *L. acerina* and *L. cotini*, the combined sequences of ITS, *TEF1-α*, *TUB2* and *RPB2* for all strains were used for the phylogenetic analysis by methods of maximum parsimony (MP), maximum likelihood (ML) and MrBayes analyses (BI) with *Diplodia mutila* and *D. seriata* as outgroups. NEXUS files were generated with Clustal X 1.83 [23] in Phylosuit v.1.2.2 [24].

ML analyses with 1000 bootstrap replicates were conducted using raxmlGUI v. 2.06 [25]. The best-fit model of nucleotide substitution for each dataset was determined using ModelFinder [26]. Topological confidence of resulted trees was assessed by maximum likelihood bootstrap proportion (MLBP) with 1000 replicates.

MP trees were generated in PAUP v.4.0b [27], using the heuristic search function with tree bisection and reconstruction as branch swapping algorithms and 1000 random addition replicates. Gaps were treated as a fifth character and the characters were unordered and given equal weight. MAXTREES were set to 5000, branches of zero length were collapsed and all multiple, equally parsimonious trees were saved. Tree length (TL), consistency index (CI), retention index (RI), rescaled consistency index (RC) and homoplasy index (HI) were calculated. Topological confidence of resulting trees was tested by maximum parsimony bootstrap proportion (MPBP) with 1000 replications, each with 10 replicates of random addition of taxa.

BI analysis was conducted by MrBayes v. 3.2.6 [28] with Markov Chain Monte Carlo algorithm. Nucleotide substitution models were determined by ModelFinder and GTR + I+G + F was estimated as the best-fit model. Two MCMC chains were run from random trees for 2,000,000 generations and sampled every 100 generations. The first 2500 trees were discarded as the burn-in phase of the analyses, and Bayesian inference posterior probability (BIPP) was determined from the remaining trees. Trees were visualized in FigTree v1.4.4.

## 3. Results

### 3.1. Phylogenetic Analyses

The combined ITS, *TEF1-α*, *TUB2* and *RPB2* data set comprised 74 taxa with *D. mutila* and *D. seriata* as the outgroups. The MP dataset consisted of 1823 characters, of which 1358 characters were constant, 115 characters were parsimony informative and 366 variable characters were parsimony uninformative. A total of 284 most-parsimonious trees with the same topology were generated, one of them is shown in Figure 1 (tree length = 1075, CI = 0.5563, RI = 0.8692, RC = 0.4835, HI = 0.4437). In the ML analyses, GTRGAMMA was specified as the model. The best scoring RAxML tree with the final ML optimization likelihood value of −8913.786383 (ln) yielded. Estimated base frequencies were as follows: A = 0.224747, C = 0.283918, G = 0.271555, T = 0.219781; substitution rates AC = 0.836915, AG = 3.800207, AT = 1.307148, CG = 1.119223, CT = 6.358526, GT = 1.000000; gamma distribution shape parameter α = 0.220772. The ML, MP and BI methods for phylogenetic analyses resulted in trees with similar topologies.

Among all the strains, 141 represented 76 *Lasiodiplodia* spp. clustered together with high support (MPBP/MLBP/BIPP = 100%/100%/1). Three isolates (JZBHD 1902, 1904 and 1905) representing *L. acerina* and three isolates (JZBPG 1901, 1903 and 1905) representing *L. cotini* clustered as distinct lineages from other *Lasiodiplodia* spp., with the support values MPBP/BIPP = 85%/0.84 and MPBP/MLBP/BIPP = 98%/100%/1, respectively. They showed a close phylogenetic relationship, respectively, with *L. henanica* and *L. citricola*.

### 3.2. Taxonomy

*Lasiodiplodia acerina* G. H. Qiao & W.T. Qin, sp. nov. MB845417; Figure 2.

Etymology: The specific epithet is in reference to the host, *Acer truncatum*, from which the fungus was isolated.

Typification: China, Beijing, Haidian district, Summer Palace, Longevity Hill, from blighted stems of *Acer truncatum*, 18 September 2019, G. H. Qiao (Holotype: JZBHDT1904, ex-type isolate: JZBHD1904).

DNA barcodes: ITS = OP117391, *TUB2* = OP141783, *RPB2* = OP141788, *TEF1-α* = OP141777.

Conidiomata were semi-immersed or superficial stromatic on PDA within 14 d, and were solitary, smooth, globose, dark grey to black, covered by dark gray mycelia without conspicuous ostioles and up to 2525 µm in diameter. Paraphyses were filiform, cylindrical, aseptate, thin-walled, hyaline, apex rounded, occasionally swollen at the base and unbranched, arising from the conidiogenous layer, extending above the level of developing conidia, and were up to 39.4 µm long and 3.0 µm wide. Conidiophores were reduced to conidiogenous cells. Conidiogenous cells were hyaline, holoblastic, smooth, discrete, thin-walled, and were cylindrical to ampulliform. Conidia were initially hyaline, ovoid to cylindrical, with a 1-µm-thick wall, (21.64-)21.97–30.83(-30.96) × (10.61-)11.48–15.87(-16.72) µm (*n* = 50, av. = 26.9 µm × 13.5 µm, L/W ratio = 2.0, by range from 1.58 to 2.61. Mature conidia turned brown with a median septum and longitudinal striations and sometimes with one vacuole. The sexual stage and spermatia were not observed.

Culture characteristics: Colonies on PDA were initially white with thick aerial mycelia reaching the lid of the plate. After 7 d colonies were fluffy, grey to black, with reverse side of the colonies black. The colonies radius reached 32 mm on PDA after 24 h, and mycelia entirely covered the surface of the plate after 48 h in darkness at 25 °C. Aerial mycelia on MEA was moderately dense and reached the lid of the plate and became olive gray to black on the surface of the plate after 7 d. The colonies radius reached 30 mm after 24 h, and 76 mm after 48 h on MEA in darkness at 25 °C. Aerial mycelia on SNA were sparse, white. The colonies radius reached 22 mm after 24 h, and 58 mm after 48 h in darkness at 25 °C. Mycelia entirely covered the surface of the plate after 72 h on all the three culture media in darkness at 25 °C.

Additional strains examined: China, Beijing, Haidian district, Summer Palace, Longevity Hill, 39.91 °N 116.41 °E, from blighted stems of *Acer truncatum*, 18 September 2019, G. H. Qiao, HDyhy1902, JZBQHD1902; *ibid.*, HDyhy1905, JZBQ1905.

Notes: Phylogenetically, as a separated linage, three strains of *L. acerina* formed sister groups with *L. henanica* (MPBP = 99%) and *L. huangyanensis* (MPBP/BIPP = 99%/0.86). Compared with the sequences of *TEF1-α* for *L. acerina*, they shared low similarities with *L. henanica* (97.71%), *L. huangyanensis* CGMCC 3.20380 (96.08%) and *L. huangyanensis* CGMCC 3.20381 (96.41%) by 7, 12 and 11 bp divergent among 306 bp, respectively. Morphologically, mycelia of *L. acerina* on MEA grew faster than that of *L. henanica* (colony radius reached 26 mm on MEA after 24 h, and more than 65 mm after 48 h in darkness at 28 °C). The length of paraphysis were longer in *L. henanica* (105 μm) [6] and *L. huangyanensis* (82 μm) [9]. In addition, *L. henanica* had smaller conidiomata (520 µm) (Table 2), and vacuoles in the conidia, which were also different from *L. acerina* [6].

Lasiodiplodia cotini G. H. Qiao & W.T. Qin, sp. nov. MB845418; Figure 3.

Etymology: The specific epithet is in reference to the host, Cotinus coggygria, from which the fungus is isolated.

Typification: China, Beijing, Pinggu district, Huangsongyu Town, Dadonggou village, from blighted stems of Cotinus coggygria, 20 October 2018, G. H. Qiao (ex-type strain: JZBPG 1905).

DNA barcodes: ITS = OP117389, TUB2 = OP141781, RPB2 = OP141787, TEF1-α = OP141775.

Conidiomata were semi-immersed or superficial stromatic, produced on PDA within 14 d, solitary, smooth, globose, dark grey to black, covered by dark gray mycelia without a conspicuous ostiole, up to 415 µm in diameter. Paraphyses arise from the conidiogenous layer, filiform, extending above the level of developing conidia, up to 41.9 µm long and 2.6 µm wide, hyaline, cylindrical, aseptate, thin-walled, apex rounded, occasionally swollen at the base and unbranched. Conidiophores were reduced to conidiogenous cells. Conidiogenous cells were hyaline, cylindrical to ampulliform, holoblastic, discrete, thin-walled and smooth. Conidia were initially hyaline, ovoid to cylindrical, with a 1-µm-thick wall, mature conidia turned brown with a median septum and longitudinal striations and sometimes with one vacuole, (19.38-)20−27(-28.81) × (12.51-)13.61−16.55(-16.62) µm (n = 50, av. = 24.28 µm × 15.4 µm, L/W ratio = 1.58, by range from 1.40 to 1.69. The sexual stage and spermatia were not observed.

Culture characteristics: Aerial mycelia on PDA were abundant, smoke-grey to olivaceous-grey with the colonies dark black on the reverse side of the plate after 7 d. The colonies radius reached 45 mm on PDA after 24 h, and mycelia entirely covered the surface of the plate after 48 h in darkness at 25 °C. The colonies radius reached 24 mm on MEA after 24 h in darkness at 25 °C, and 51 mm after 48 h. Aerial mycelium is moderately dense and grey. The colonies radius reached 14 mm on SNA after 24 h, and 43 mm after 48 h in darkness at 25 °C. Aerial mycelium on SNA is sparse and white. After 72 h mycelia entirely covered the surface of the plates of the three culture media.

Additional strains examined: China, Beijing, Pinggu district, Huangsongyu Town, Dadonggou village, 40.23 °N 117.29 °E from blighted stems of Cotinus coggygria, 20 October 2018, G. H. Qiao, PGhsy 1901, JZBPG1901; ibid., PGhsy 1903, JZBPG1903.

Notes: Phylogenetically, three strains of L. cotini clustered together (MPBP/MLBP/BIPP = 98%/100%/1) and are closely related to L. citricola (MPBP/MLBP/BIPP = 68%/93%/0.94). Comparison of the sequence data indicated that they shared 4 bp divergent among 259 bp for TEF1-α (98.46%). Morphologically, the colonies of L. citricola and L. cotini were not obviously different; however, L. cotini has smaller paraphyses than those of L. citricola (125 × 3–4 μm) [29] and L. cinnamomi (106 × 3–4 μm) [30]. In addition, larger conidia of L. cinnamomi (18.7–21.1 × 12.7–14.1 μm) also make it distinguishable from L. cotini (Table 2) [30].

## 4. Discussion

To explore the taxonomic positions of the genus Lasiodiplodia, the phylogenetic tree was constructed based on the combined sequences of ITS, TEF1-α, TUB2 and RPB2 with D. mutila and D. seriata used as outgroups. Two novel species, L. acerina and L. cotini, were found based on the integrated studies of phenotypic and molecular data. All investigated Lasiodiplodia species clustered together (Figure 1), which was basically congruent with the results of a previous study [6]. Lasiodiplodia acerina and L. cotini clustered as separated terminal branches at the top of the tree, and were closely related to L. henanica [6] and L. citricola [30], respectively, but they differed from each other in characters of conidiomata, conidia and paraphyses, etc. (Figure 2 and Figure 3; Table 2).

Although many species in Lasiodiplodia were differentiated on the basis of morphological characters, it is necessary to combine the morphology and molecular data for definitive identifications. The phylogenetic tree in this study was comprised of 76 Lasiodiplodia species represented by 141 strains. When our two new species joined, the tree topology was somewhat changed, including the relationships among species. Lasiodiplodia acerina and four newly reported species, L. henanica on blueberries [6], L. huangyanensis and L. ponkanicola on citrus [9], and L. cinnamomic on Cinnamomum camphora in China formed a separated terminal branch [29]. Lasiodiplodia citricola was reported as the sister group of L. paraphysoides and L. aquilariae [6,9]; however, in this study, four strains representing L. citricola were closely related to L. cotini represented by our three strains (MPBP/MLBP/BIPP = 56%/64%/0.8). Lasiodiplodia citricola were far away from L. paraphysoides, a novel species reported on blueberries [6] as a result of L. cotini in our study and L. mitidjana on citrus [30] joining in the phylogenetic tree.

Further analysis showed that the Lasiodiplodia species sampling from China tend to cluster together (Table 1 and Figure 1), which may be the result of the comprehensive action of fungal adaptive ability, regional climate and human-mediated factors. For example, five newly reported species sampling from China in recent years, L. acerina, L. cinnamomi, L. henanica, L. huangyanensis and L. ponkanicola formed a high-supported group (MPBP/BIPP = 99%/0.68). Geographically, species in the genus Lasiodiplodia tend to live in tropical or subtropical areas or in warm temperature areas associated with stem diseases of woody substrates [30,31]. In this study, two newly described species of Lasiodiplodia were also isolated from the blighted stem of A. truncatum and C. coggygria in Beijing, which are distributed in subtropical or warm temperate areas in China (Table 1).

Acer truncatum and Cotinus coggygria are two kinds of landscape trees that play important roles in urban greening construction. Botryosphaeria dothidea, Fusarium oxysporum, Neofusicoccum parvum and Pestalotiopsis microspora have been reported to be associated with diseased leaves and stems of Acer spp. [32,33,34,35], and Alternaria alternata, Botryosphaeria dothidea and Verticillum dahlia have been isolated from diseased leaves and stems of C. coggygria [36,37,38]; to our knowledge, this is the first report of Lasiodiplodia being associated with A. truncatum and C. coggygria.

Along with an increasing number of species recognized in the genus Lasiodiplodia, our understanding of the genus will become more sophisticated and intelligible through the integrated studies on morphology and phylogeny. Accumulations of our knowledge on Lasiodiplodia will provide useful information for establishing reasonable species concepts, and understand co-relations between morphology and sequence data in the future, which will lay further foundations for the scientific management of stem blight diseases and improvement in the landscape effect in the process of urban greening construction.

## 5. Conclusions

This study recognized two novel *Lasiodiplodia* species from blighted stems of *A. truncatum* and *C. coggygria*, which were the first reports of *Lasiodiplodia* associated with these two horticulture trees in China. The discovery provided a better understanding of the biodiversity and phylogeny of the genus *Lasiodiplodia* and is beneficial for future evaluation of the potential usages and functions of the new species.

## Figures and Tables

**Figure 1 biology-11-01459-f001:**
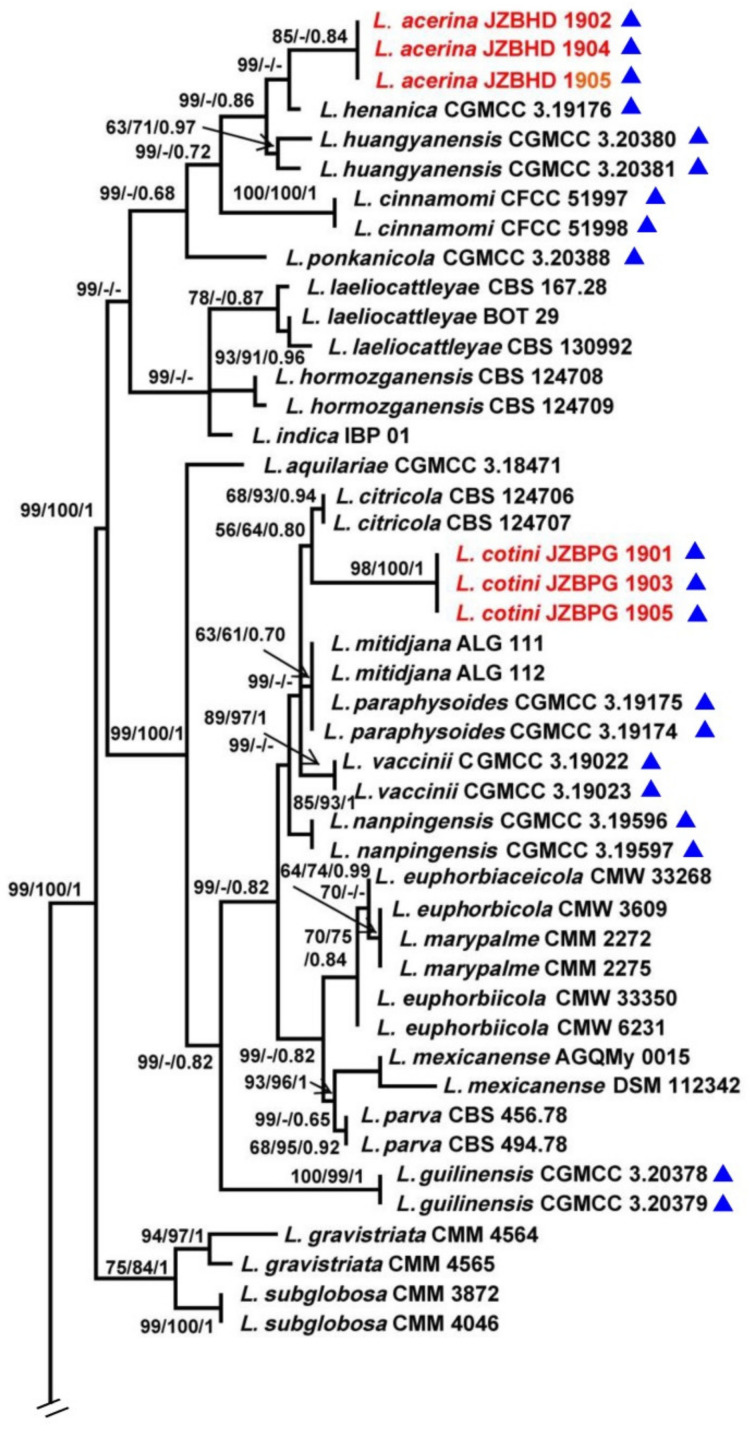
Maximum parsimony phylogram reconstructed from the combined sequences of ITS, *TEF1-α*, *TUB2* and *RPB2* of *Lasiodiplodia*. MPBP above 50% (**left**), MLBP above 50% (**middle**), BIPP above 0.7 (**right**) are indicated at the nodes. New species proposed are indicated in red font. The tree is rooted to *Diplodia mutila* and *D. seriata*. The strains isolated from samples of China are marked in blue triangles.

**Figure 2 biology-11-01459-f002:**
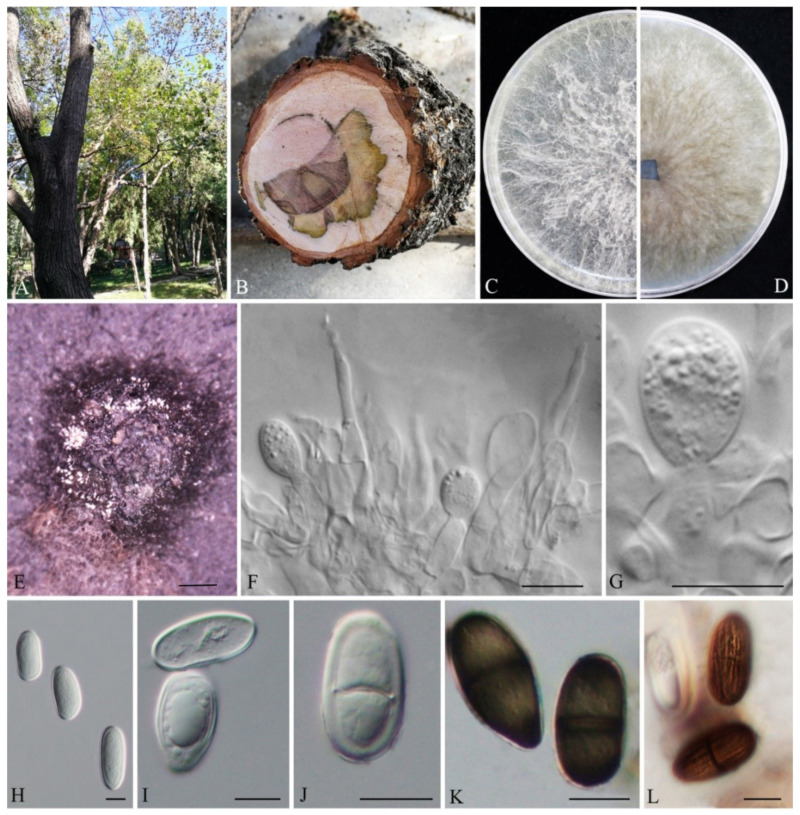
*Lasiodiplodia acerina* (JZBHD 1904). (**A**) Disease tree in the field. (**B**) Cross-section of stem. (**C**,**D**) Culture grown on PDA. (**E**) Conidiomata developing on PDA. (**F**,**G**) Conidia developing on conidiogenous cells between paraphyses. (**H**–**L**) Conidia. Scale bars: E = 200 μm, F−L = 10 μm.

**Figure 3 biology-11-01459-f003:**
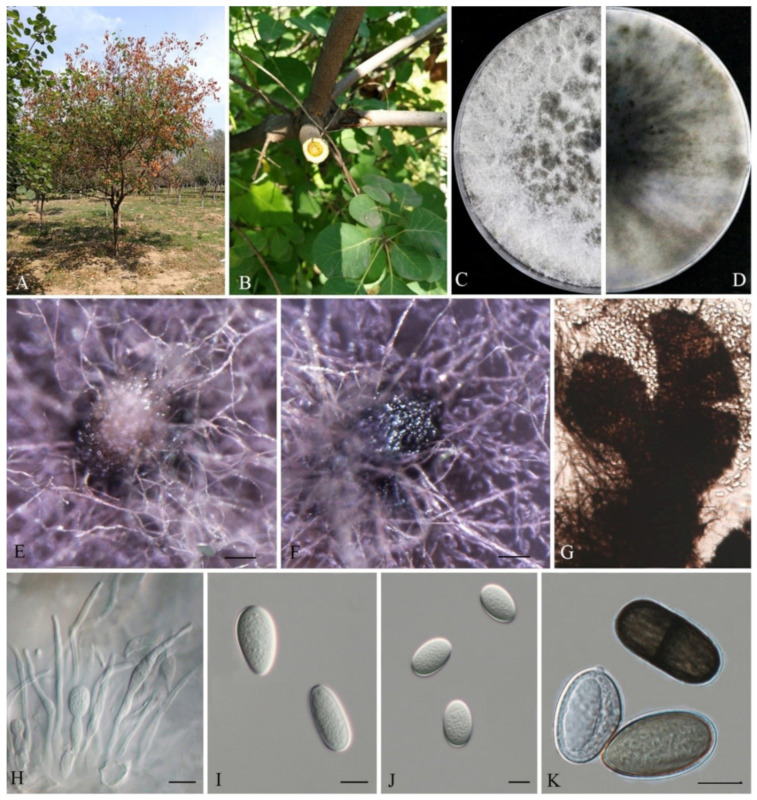
Lasiodiplodia cotini (JZBPG 1905). (**A**) Diseased tree in the field. (**B**) Cross-section of the blighted stem. (**C**,**D**) Culture grown on PDA. (**E**,**F**) Conidiomata developing on PDA. (**G**) Crushed conidiomata with many conidia. (**H**) Conidia developing on conidiogenous cells between paraphyses. (**I**–**K**) Conidia. Scale bars: E − F = 100 μm, H − K = 10 μm.

**Table 1 biology-11-01459-t001:** Details of *Lasiodiplodia* strains investigated in this study.

Species	Strain	Host	Locality	GenBank Accession Numbers			
				ITS	*TEF1-α*	*TUB*	*RPB2*
*Lasiodiplodia acaciae*	CBS 136434T	*Acacia* sp.	Indonesia	MT587421	MT592133	MT592613	MT592307
*L. acerina*	JZBHD1902	*Acer truncatum*	China	**OP117390**	**OP141776**	**OP141782**	**N/A**
*L. acerina*	JZBHD1904T	*Acer truncatum*	China	**OP117391**	**OP141777**	**OP141783**	**OP141788**
*L. acerina*	JZBHD1905	*Acer truncatum*	China	**OP117392**	**OP141778**	**OP141784**	**OP141789**
*L.americana*	CERC1962	*Pistacia vera*	USA	KP217060	KP217068	KP217076	N/A
*L.americana*	CERC1961T	*Pistacia vera*	USA	KP217059	KP217067	KP217075	N/A
*L.americana*	CERC1960	*Pistacia vera*	USA	KP217058	KP217066	KP217074	N/A
*L. aquilariae*	CGMCC 3.18471T	*Aquilaria crassna*	Laos	KY783442	KY848600	N/A	KY848562
*L. avicenniae*	CMW 41467T	*Avicennia marina*	South Africa	KP860835	KP860680	KP860758	KU587878
*L. avicenniae*	LAS 199	*Avicennia marina*	South Africa	KU587957	KU587947	KU587868	KU587880
*L. avicenniarum*	MFLUCC 17-2591T	*Avicennia marina*	Thailand	MK347777	MK340867	N/A	N/A
*L. brasiliense*	CMW 35884	*Adansonia* sp.	Laos	KU887094	KU886972	KU887466	KU696345
*L. brasiliense*	CBS 115447	*Psychotria tutcheri*	China	MT587422	MT592134	MT592614	MT592308
*L. brasiliensis*	CMM 4015T	*Mangifera indica*	Brazil	JX464063	JX464049	N/A	N/A
*L. brasiliensis*	CMM 4469	*Anacardium occidentale*	Brazil	KT325574	KT325580	N/A	N/A
*L. bruguierae*	CMW 41470T	*Bruguiera gymnorrhiza*	South Africa	KP860833	KP860678	KP860756	KU587875
*L. bruguierae*	CMW 42480	*Bruguiera gymnorrhiza*	South Africa	KP860832	KP860677	KP860755	KU587876
*L. caatinguensis*	CMM 1325T	*Citrus sinensis*	Brazil	KT154760	KT008006	KT154767	N/A
*L. caatinguensis*	IBL 381	*Spondias purpurea*	Brazil	KT154757	KT154751	KT154764	N/A
*L. chiangraiensis*	MFLUCC 21-0003T	*/*	Thailand	MW760854	MW815630	MW815628	N/A
*L. chiangraiensis*	GZCC 21-0003	*/*	Thailand	MW760853	MW815629	MW815627	N/A
*L. chinensis*	CGMCC 3.18061T	*/*	China	KX499889	KX499927	KX500002	KX499965
*L. chinensis*	CGMCC 3.18063	*Canarium parvum*	China	KX499891	KX499929	KX500004	KX499967
*L. chonburiensis*	MFLUCC 16-0376T	*Pandanaceae*	Thailand	MH275066	MH412773	MH412742	N/A
*L. cinnamomi*	CFCC 51997T	*Cinnamomum camphora*	China	MG866028	MH236799	MH236797	MH236801
*L. cinnamomi*	CFCC 51998	*Cinnamomum camphora*	China	MG866029	MH236800	MH236798	MH236802
*L. citricola*	CBS 124707T	*Citrus* sp.	Iran	GU945354	GU945340	KU887505	KU696351
*L. citricola*	CBS 124706	*Citrus* sp.	Iran	GU945353	GU945339	KU887504	KU696350
*L. clavispora*	CGMCC 3.19594T	*Vaccinium uliginosum*	China	MK802166	N/A	MK816339	MK809507
*L. clavispora*	CGMCC 3.19595	*Vaccinium uliginosum*	China	MK802165	N/A	MK816338	MK809506
*L. cotini*	JZBPG1901	*Cotinus coggygria*	China	**OP117387**	**OP141773**	**OP141779**	**OP141785**
*L. cotini*	JZBPG1903	*Cotinus coggygria*	China	**OP117388**	**OP141774**	**OP141780**	**OP141786**
*L. cotini*	JZBPG1905T	*Cotinus coggygria*	China	**OP117389**	**OP141775**	**OP141781**	**OP141787**
*L. crassispora*	CBS 118741T	*Santalum album*	Australia	DQ103550	DQ103557	KU887506	KU696353
*L. crassispora*	CMW 13488	*Eucalyptus urophylla*	Venezuela	DQ103552	DQ103559	KU887507	KU696352
*L. crassispora*	WAC 12533	*Santalum album*	Australia	DQ103550	DQ103557	KU887506	KU696353
*L. curvata*	CGMCC 3.18456T	*Aquilaria crassna*	Laos	KY783437	KY848596	KY848529	KY848557
*L. curvata*	CGMCC 3.18476	*Aquilaria crassna*	Laos	KY783443	KY848601	KY848532	KY848563
*L. endophytica*	MFLUCC 18-1121T	*Magnolia acuminata*	China	MK501838	MK584572	MK550606	N/A
*L. euphorbicola*	CMW 3609T	*Jatropha curcas*	Brazil	KF234543	KF226689	KF254926	N/A
*L. euphorbiicola*	CMW 33350T	*Adansonia digitata*	Botswana	KU887149	KU887026	KU887455	KU696346
*L. euphorbiicola*	CMW 36231	*Adansonia digitata*	Zimbabwe	KU887187	KU887063	KU887494	KU696347
*L. euphorbiaceicola*	CMW 33268T	*Adansonia* sp.	Senegal	KU887131	KU887008	KU887430	KU887367
*L. exigua*	BL184T	*Retama raetam*	Tunisia	KJ638318	KJ638337	N/A	N/A
*L. exigua*	CBS 137785	*Retama raetam*	Tunisia	KJ638317	KJ638336	KU887509	KU696355
*L. fujianensis*	CGMCC 3.19593T	*Vaccinium uliginosum*	China	MK802164	MK887178	MK816337	MK809505
*L. gilanensis*	CBS 124704T	*Citrus* sp.	Iran	GU945351	GU945342	KU887511	KU696357
*L. gilanensis*	CBS 124705	*Citrus* sp.	Iran	GU945352	GU945341	KU887510	KU696356
*L. gonubiensis*	CMW 14077T	*Syzygium cordatum*	South Africa	AY639595	DQ103566	DQ458860	KU696359
*L. gonubiensis*	CMW 14078	*Syzygium cordatum*	South Africa	AY639594	DQ103567	EU673126	KU696358
*L. gravistriata*	CMM 4564T	*Anacardium humile*	Brazil	KT250949	KT250950	N/A	N/A
*L. gravistriata*	CMM 4565	*Anacardium humile*	Brazil	KT250947	KT266812	N/A	N/A
*L. guilinensis*	CGMCC3.20378T	*Citrus sinensis*	China	MW880672	MW884175	MW884204	MW884149
*L. guilinensis*	CGMCC3.20379	*Citrus unshiu*	China	MW880673	MW884176	MW884205	MW884150
*L. henanica*	CGMCC3.19176T	*Vaccinium uliginosum*	China	MH729351	MH729357	MH729360	MH729354
*L. hormozganensis*	CBS 124709T	*Olea* sp.	Iran	GU945355	GU945343	KU887515	KU696361
*L. hormozganensis*	CBS 124708	*Mangifera indica*	Iran	GU945356	GU945344	KU887514	KU696360
*L. huangyanensis*	CGMCC 3.20380T	*Citrus lata*	China	MW880674	MW884177	MW884206	MW884151
*L. huangyanensis*	CGMCC 3.20381	*Citrus unshiu*	China	MW880675	MW884178	MW884207	MW884152
*L.hyalina*	CGMCC 3.17975T	*Acacia confusa*	China	KX499879	KX499917	KX499992	KX499955
*L. hyalina*	CGMCC 3.18383	*/*	China	KY767661	KY751302	KY751299	KY751296
*L. indica*	IBP 01T	*angiospermic wood*	India	KM376151	N/A	N/A	N/A
*L. iranensis*	CBS 124710T	*Salvadora persica*	Iran	GU945348	GU945336	KU887516	KU696363
*L. iranensis*	CBS 124711	*Juglans* sp.	Iran	GU945347	GU945335	KU887517	KU696362
*L. irregularis*	CGMCC3.18468T	*Aquilaria crassna*	Laos	KY783472	KY848610	KY848553	KY848592
*L. jatrophicola*	CMM 3610T	*Jatropha curcas*	Brazil	KF234544	KF226690	KF254927	N/A
*L.jatrophicola*	CMW36237	*Adansonia* sp.	Brazil	KU887121	KU886998	KU887499	KU696348
*L.jatrophicola*	CMW36239	*Adansonia* sp.	Brazil	KU887123	KU887000	KU887501	KU696349
*L. krabiensis*	MFLUCC 17-2617T	*Bruguiera* sp.	Thailand	MN047093	MN077070	N/A	N/A
*L. laeliocattleyae*	CBS 130992T	*Mangifera indica*	Egypt	KU507487	KU507454	KU887508	KU696354
*L. laeliocattleyae*	BOT 29	*Mangifera indica*	Egypt	JN814401	JN814428	N/A	N/A
*L. laeliocattleyae*	CBS 167.28	*Laeliocattleya* sp.	Italy	KU507487	KU507454	MT592618	MT592313
*L. laosensis*	CGMCC 3.18464T	*Aquilaria crassna*	Laos	KY783471	KY848609	KY848552	KY848591
*L. laosensis*	CGMCC 3.18473	*Aquilaria crassna*	Laos	KY783450	KY848603	KY848536	KY848570
*L. lignicola*	CBS 134112T	*/*	Thailand	JX646797	KU887003	JX646845	KU696364
*L. lignicola*	MFLUCC 11-0435	*/*	Thailand	JX646797	JX646862	JX646845	KP872470
*L. lignicola*	MFLUCC 11-0656	*/*	Thailand	JX646798	JX646863	JX646846	N/A
*L. linhaiensis*	CGMCC 3.20386T	*Citrus unshiu*	China	MW880677	MW884180	MW884209	MW884154
*L. linhaiensis*	CGMCC 3.20383	*Citrus sinensis*	China	MW880678	MW884181	MW884210	MW884155
*L. loidaceae*	DSM 112340T	*Lodoicea maldivica*	Mexico	MW274148	MW604230	MW604240	MW604219
*L. loidaceae*	DSM 112341	*Lodoicea maldivica*	Mexico	MW274146	MW604229	MW604239	MW604218
*L. macroconidia*	CGMCC 3.18479T	*Aquilaria crassna*	Laos	KY783438	KY848597	KY848530	KY848558
*L. macrospora*	CMM 3833T	*Jatropha curcas*	Brazil	KF234557	KF226718	KF254941	N/A
*L. magnoliae*	MFLUCC 18-0948T	*Magnolia acuminata*	China	MK499387	MK568537	MK521587	N/A
*L. mahajangana*	CMW 27801T	*Terminalia catappa*	Madagascar	FJ900595	FJ900641	FJ900630	KU696365
*L. mahajangana*	CMW 27818	*Terminalia catappa*	Madagascar	FJ900596	FJ900642	FJ900631	KU696366
*L. mahajangana*	CBS:125267	*Terminalia sambesiaca*	Tanzania	MT587428	MT592140	MT592622	MT592318
*L. margaritacea*	CBS 122519T	*Adansonia gibbosa*	Australia	EU144050	EU144065	KU887520	KU696367
*L. margaritacea*	CBS 138291	*Combretum obovatum*	Zambia	KP872322	KP872351	KP872381	KP872431
*L. marypalme*	CMM 2275T	*Carica papaya*	Brazil	KC484843	KC481567	N/A	N/A
*L. marypalme*	CMM 2272	*Carica papaya*	Brazil	KC484842	KC481566	N/A	N/A
*L. mediterranea*	CBS 137783T	*Quercus ilex*	Italy	KJ638312	KJ638331	KU887521	KU696368
*L. mediterranea*	CBS 137784	*Vitis vinifera*	Italy	KJ638311	KJ638330	KU887522	KU696369
*L. mexicanense*	DSM 112342T	*Chamaedorea seifrizii*	Mexico	MW274151	MW604234	MW604243	MW604222
*L. mexicanense*	AGQMy 0015	*Chamaedorea seifrizii*	Mexico	MW274150	MW604233	MW604242	MW604221
*L. microcondia*	CGMCC 3.18485T	*Aquilaria crassna*	Laos	KY783441	KY848614	N/A	KY848561
*L. missouriana*	UCD 2193MOT	*Vitis* sp.	USA	HQ288225	HQ288267	HQ288304	KU696370
*L. missouriana*	UCD 2199MO	*Vitis* sp.	USA	HQ288226	HQ288268	HQ288305	KU696371
*L. mitidjana*	ALG111T	*Citrus* sp.	Algeria	MN104115	MN159114	N/A	N/A
*L. mitidjana*	ALG112	*Citrus* sp.	Algeria	MN104116	MN159115	N/A	N/A
*L. nanpingensis*	CGMCC3.19596T	*Vaccinium uliginosum*	China	MK802167	N/A	MK816340	MK809508
*L. nanpingensis*	CGMCC3.19597	*Vaccinium uliginosum*	China	MK802168	N/A	MK816341	MK809509
*L. pandanicola*	MFLUCC 16-0265T	*Pandanaceae*	Thailand	MH275068	MH412774	MH412744	N/A
*L. pandanicola*	GBLZ 16BO-008T	*Litchi chinensis*	China	MN540679	N/A	MN539183	N/A
*L.paraphysoide*	CGMCC 3.19174T	*Vaccinium uliginosum*	China	MH729349	MH729355	MH729358	MH729352
*L.paraphysoides*	CGMCC 3.19175	*Vaccinium uliginosum*	China	MH729350	MH729356	MH729359	MH729353
*L. parva*	CBS 456.78T	/	USA	EF622083	EF622063	KU887523	KU696372
*L. parva*	CBS 494.78	*Cassava-field soil*	Colombia	EF622084	EF622064	EU673114	KU696373
*L. plurivora*	STE-U 5803T	*Prunus salicina*	South Africa	EF445362	EF445395	KP872421	KP872479
*L. plurivora*	STE-U 4583	*Vitis vinifera*	South Africa	AY343482	EF445396	KP872422	KP872480
*L. ponkanicola*	CGMCC3.20388T	*Citrus reticulata*	China	MW880685	MW884188	MW884214	MW884159
*L. pontae*	CMM 1277T	*Spondias purpurea*	Brazil	KT151794	KT151791	KT151797	N/A
*L. pontae*	CBS 117454	*Eucalyptus urophylla*	Venezuela	MT587432	MT592144	MT592626	N/A
*L. pseudotheobromae*	CBS 116459T	*Gmelina arborea*	Costa Rica	EF622077	EF622057	EU673111	KU696376
*L.pseudotheobromae*	CGMCC 3.18047	*Pteridium aquilinum*	China	KX499876	KX499914	KX499989|	KX499952
*L. pseudotheobromae*	CBS 121772	*Acacia mellifera*	Namibia	EU101310	EU101355	MT592627	MT592323
*L. pyriformis*	CBS 121770T	*Acacia mellifera*	Namibia	EU101307	EU101352	KU887527	KU696378
*L. pyriformis*	CBS 121771	*Acacia mellifera*	Namibia	EU101308	EU101353	KU887528	KU696379
*L. rubropurpurea*	WAC 12535T	*Eucalyptus grandis*	Australia	DQ103553	DQ103571	EU673136	KU696380
*L. rubropurpurea*	WAC 12536	*Eucalyptus grandis*	Australia	DQ103554	DQ103572	KU887530	KU696381
*L. sterculiae*	CBS342.78T	*Sterculia oblonga*	Germany	KX464140	KX464634	KX464908	KX463989
*L. subglobosa*	CMM 3872T	*Jatropha curcas*	Brazil	KF234558	KF226721	KF254942	N/A
*L. subglobosa*	CMM 4046	*Jatropha curcas*	Brazil	KF234560	KF226723	KF254944	N/A
*L. swieteniae*	MFLUCC 18-0244T	*Swietenia mahagoni*	Thailand	MK347789	MK340870	MK412877	N/A
*L. syzygii*	MFLUCC 19-0257T	*Syzygium samarangense*	Thailand	MT990531	MW016943	MW014331	N/A
*L. syzygii*	CBS:120512	*Syzygium samarangense*	Thailand	MT587434	MT592147	MT592632	N/A
*L. syzygii*	GUCC 9719.2	*Syzygium samarangense*	Thailand	MW081991	MW087101	MW087104	N/A
*L. tenuiconidia*	CGMCC 3.18449T	*Aquilaria crassna*	Laos	KY783466	KY848619	N/A	KY848586
*L. thailandica*	CBS 138760T	*Mangifera indica*	Thailand	KJ193637	KJ193681	N/A	N/A
*L. thailandica*	CGMCC 3.18384	*Albizia chinensis*	China	KY767663	KY751304	KY751301	KY751298
*L. thailandica*	MUCC<JPN>:2738	*Bryophyllum pinnatum*	Japan	LC567321	LC567750	LC567780	LC567810
*L. theobromae*	CBS 164.96T	/	Papua New Guinea	AY640255	AY640258	KU887532	KU696383
*L. theobromae*	CBS 111530	*Leucospermum* sp.	USA	EF622074	EF622054	KU887531	KU696382
*L. tropica*	CGMCC 3.18477T	*Aquilaria crassna*	Laos	KY783454	KY848616	KY848540	KY848574
*L. vaccinii*	CGMCC 3.19022T	*Vaccinium uliginosum*	China	MH330318	MH330327	MH330324	MH330321
*L. vaccinii*	CGMCC 3.19023	*Vaccinium uliginosum*	China	MH330319	MH330329	MH330326	MH330322
*L. venezuelensis*	WAC 12539T	*Acacia mangium*	Venezuela	DQ103547	DQ103568	KU887533	KP872490
*L. venezuelensis*	WAC 12540	*Acacia mangium*	Venezuela	DQ103548	DQ103569	KU887534	KP872491
*L. viticola*	CBS 128313T	*Vitis vinifera*	USA	HQ288227	HQ288269	HQ288306	KU696385
*L. viticola*	UCD 2604MO	*Vitis vinifera*	USA	HQ288228	HQ288270	HQ288307	KU696386
*L. vitis*	CBS 124060T	*Vitis vinifera*	Italy	KX464148	KX464642	KX464917	KX463994
*Diplodia mutila*	CMW 7060T	*Fraxinus excelsior*	Netherlands	AY236955	AY236904	AY236933	EU339574
*D. seriata*	CBS 112555T	*Vitis vinifera*	Portugal	AY259094	AY573220	DQ458856	N/A

T: Type collections. N/A: no sequences in GenBank. /: unknown host. Numbers in bold indicate newly submitted sequences in this study.

**Table 2 biology-11-01459-t002:** Morphological characteristic comparison between *L. acerina*, *L. cotini* and their close relatives.

Species	Length of Conidia (μm)	Width of Conidia (μm)	Average L/W of Conidia	L/W Range of Conidia	Length of Paraphyses (μm)	Width of Paraphyses (μm)	Size of Conidiomata (μm)	Reference
*L. acerina*	(21.64-)21.97–30.83 (-30.96)	(10.61-)11.48–15.87(-16.72)	2.00	1.58–2.61	39.4	3	2525	This study
*L. henanica*	(14-)19–26(-27)	10–13 (-15)	1.86	1.17–2.60	105	4	520	[6]
*L. huangyanensis*	(21-)28–32.5(-34)	(13-)14–16(-17)	2.00	-	82	3–4	-	[9]
*L. cinnamomi*	(17.5-)18.7–21.1(-22.4)	(11.5-)12.7–14.1(-15.5)	1.50	-	106	3–4	-	[29]
*L. citricola*	(20-)22–27(-31)	(10.9-)12–17(-19)	1.60	-	125	3–4	-	[30]
*L. cotini*	(19.38-)20–27(-28.81)	(12.51-)13.61–16.55(-16.62)	1.58	1.40–1.69	41.9	2.6	415	This study

## Data Availability

All data analyzed in this study are curated from the public domain.

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
