# Peer review of "Two Novel Lasiodiplodia Species from Blighted Stems of Acer truncatum and Cotinus coggygria in China"

_biology, 2022, doi:10.3390/biology11101459_

Round 1

Reviewer 1 Report

The manuscript is high quality and I recommend it to be published in Biology. Minor revision is:

1.Gene name for protein should be italic.

      2.Needs a scale bar in photograph for conidiomata.

Author Response

Responses to Reviewer 1 Comments

Point 1: Gene name for protein should be italic.

Response 1: Thank you for your suggestion and the gene names have been indicated italic in the revised manuscript.

Point 2: Needs a scale bar in photograph for conidiomata.

Response 2: Thank you for your suggestion. The scale bars of conidiomata have been added to figure 2 and figure 3 in the revised manuscript.

Reviewer 2 Report

The article is very well constructed. A useful article based on up-to-date information. Publishing with minor corrections is fine for me.

Author Response

Responses to Reviewer 2 Comments

The article is very well constructed. A useful article based on up-to-date information. Publishing with minor corrections is fine for me.

Thank you for your suggestion.

Reviewer 3 Report

This manuscript "Two Novel Lasiodiplodia Species from Blighted Stems of Acer truncatum and Cotinus coggygria in China" is an interesting piece of work but it needs some revisions before it is considered. The authors are encouraged to do the koch's postulate for the two patghogens. The comments and suggestions are annotated in the PDF. 

Author Response

Responses to Reviewer 3 Comments

Point 1: Authors should do koch's postulates to confirm the pathogenicity.

Response 1: Thank you for your suggestion. Pathogenicity is necessary for the causal agent to fullfill the koch's postulates, however, this study focused on defining two new species isolated from blighted stems of Acer truncatum and Cotinus coggygria by integrated analyses of phenotypic features, culture characteristics and molecular analyses. It is speculated that they may be related to the blight disease of these two plants, but whether they are pathogenic or not deserves further research.

Point 2: don't repeat the words in the title as key words.

Response 2: Thank you for your suggestion and the repeat word has been deleted.

Point 3: If you add authorities, you need to add all taxa when you mention for the first time.

Response 3: Thank you for your suggestion and the authorities has been deleted.

Point 4: “Although many species in Lasiodiplodia were differentiated on the basis of morphological characters, it is necessary to combine the morphology and molecular data for definitive identifications ”Move to discussion section.

Response 4: Thank you for your suggestion and the sentences has been moved to the begin of the second paragraph of discussion section.

Point 5: ”internal transcribed spacer region (ITS), translation elongation factor 1-α (TEF1-α), beta-tubulin (TUB2) and RNA polymerase II subunit b (RPB2) genes” Only use abbreviations as the full name already mentioned in abstract.

Response 5: Thank you for your suggestion and the full names in the third paragraph have been deleted.

Point 6: “In view of the questionable status of several species in Lasiodiplodia, there is an urgent need to reassess all of the species currently accepted in this genus.” Likely discussion part

Response 6: Thank you for your suggestion and the sentences has been deleted.

Point 7: Where did you deposit the herbarium?

Response 7: “Specimens were deposited in the culture collection of Institute of Plant Protection, Beijing Academy of Agriculture and Forestry Sciences. ”, which has been added to “2.1. Isolates and specimens” in the revised copy.

Point 8: Add final ML optimization and model information results

Response 8: Thank you for your suggestion and the relative parameters “In the ML analyses, GTRGAMMA was specified as the model. The best scoring RAxML tree with the final ML optimization likelihood value of -8913.786383 (ln) yielded. Estimated base frequencies were as follows: A =0.224747, C =0.283918, G = 0.271555, T =0.219781; substitution rates AC = 0.836915, AG = 3.800207, AT = 1.307148, CG = 1.119223, CT = 6.358526, GT = 1.000000; gamma distribution shape parameter α = 0.220772.” have been added to the revised copy.

Point 9: “DNA barcodes: ITS = OP117391, TUB2 = OP141783, RPB2 = OP141788, TEF1-α = 220 OP141777.”Should be in Table. 1

Response 9: Thank you for your suggestion. These DNA barcodes indeed were showed in Table 1, but according to the new specise published in Journal of Fungi, these barcodes of type speceis generally indicated here again.

Point 10: “The sexual stage was not observed. Conidiomata were stromatic on PDA within 14... ...”Carefully check and check how they write descriptions in academic paper

Response 10: Thank you for your suggestion. We carefully recheck the description of Lasiodiplodia and rewrite the description the characteristics of our two new species in the revised copy.

Lasiodiplodia acerina-“Conidiomata were semi-immersed or superficial stromatic on PDA within 14 d, and were solitary, smooth, globose, dark grey to black, covered by dark gray mycelia without conspicuous ostioles and up to 2525 µm in diameter. Paraphyses were filiform, cylindrical, aseptate, thin-walled, hyaline, apex rounded, occasionally swollen at the base and unbranched, arising from the conidiogenous layer, extending above the level of developing conidia, and were up to 39.4 µm long and 3.0 µm wide. Conidiophores were reduced to conidiogenous cells. Conidiogenous cells were hyaline, holoblastic, smooth,  discrete, thin-walled, and were cylindrical to ampulliform. Conidia were initially hyaline, ovoid to cylindrical, with a 1 µm thick wall,  (21.64-)21.97-30.83(-30.96)×(10.61-)11.48-15.87(-16.72) µm (n=50, av.=26.9×13.5 µm, L/W ratio=2.0, by range from 1.58 to 2.61. Mature conidia turned brown with a median septum and longitudinal striations and sometimes with one vacuole. The sexual stage and spermatia were not observed.

Lasiodiplodia cotini-“Conidiomata were semi-immersed or superficial stromatic, produced on PDA within 14 d, solitary, smooth, globose, dark grey to black, covered by dark gray mycelia without a conspicuous ostiole, up to 415 µm in diameter. Paraphyses arise from the conidiogenous layer, filiform, extending above the level of developing conidia, up to 41.9 µm long and 2.6 µm wide, hyaline, cylindrical, aseptate, thin-walled, apex rounded, occasionally swollen at the base and unbranched. Conidiophores were reduced to conidiogenous cells. Conidiogenous cells were hyaline, cylindrical to ampulliform, holoblastic, discrete, thin-walled and smooth. Conidia initially hyaline, ovoid to cylindrical, with a 1 µm thick wall, mature conidia turned brown with a median septum and longitudinal striations and sometimes with one vacuole, (19.38-)20-27(-28.81)×(12.51-)13.61-16.55(-16.62) µm (n= 50, av.=24.28×15.4 µm, L/W ratio=1.58, by range from 1.40 to 1.69. The sexual stage and spermatia were not observed. “

Point 11: “Notes”Add Blast search results

Response 11: Thank you for your suggestion. The Blast results were respectively displayed in “Compared with the sequences of TEF1-α for L. acerina, they shared low similarities with L. henanica (97.71%), L. huangyanensis CGMCC 3.20380 (96.08%) and L. huangyanensis CGMCC 3.20381 (96.41%) by 7, 12 and 11bp divergent among 306 bp, respectively.” for L. acerina, and “Comparison of the sequence data indicated that they shared 4 bp divergent among 259 bp for TEF1-α (98.46%)” for L. cotini.

Point 12: “Discussion part”Please check any fungi have been reported on Acer truncatum and Cotinus coggygria

Response 12: Thank you for your suggestion. We have check the fungi reported on Acer truncatum and Cotinus coggygria, and found that Fusarium oxysporum and Phyllactinia corylea has been reported to be associated with A. mono, Pestalotiopsis microspora associated with A. rubrum, Botryosphaeria dothidea has been reported associated with A. platanoides, Neofusicoccum parvum has been reported to be associated with bark canker of A. pseudoplatanus[1-4]. Alternaria alternata, Botryosphaeria dothidea, Uncinula vernieiferae and Verticillum dahlia have been reported to be associated with C. coggygria[5-7]. To our knowledge, this is the first report of Lasiodiplodia associated with blighted stem of A. truncatum and C. coggygria.

  1. Zhao, X., Li, H., Zhou, L., Chen, F., Chen, F. Wilt of Acer negundo caused by Fusarium nirenbergiae in China. Journal of Forestry Research 2020, 31(5): 2013-2022, doi: 10.1007/s11676-019-00996-9.
  2. Cui, C., Wang, Y., Jiang, J., Hui, O., Qin, S., Huang, T. Identification of the pathogen causing brown spot disease of 'October Glory'. Scientia Silvae Sinicae 2015, 51(10):142-147, doi: 10.11707 /j.1001-7488.20151018.
  3. Moricca, S.; Uccello, A.; Ginetti, B.; Ragazzi, A. First Report of Neofusicoccum parvum Associated with Bark Canker and Dieback of Acer pseudoplatanus and Quercus robur in Italy. Plant Disease 2012, 96, 1699-1699, doi:10.1094/pdis-06-12-0543-pdn.
  4. Wang, X.; Li, Y.X.; Dong, H.X.; Jia, X.Z.; Zhang, X.Y. First report of Botryosphaeria dothidea causing canker of Acer platanoides in China. Plant Disease 2015, 99, 1857-1857, doi:10.1094/pdis-03-15-0265-pdn
  5. Zhang, S., Liang, W., Yang, Q. First report of Alternaria alternata causing leaf spot on Cotinus coggygria in China. Plant Disease 2018, 102(12) , doi: 10.1094/PDIS-05-18-0718-PDN.
  6. Xiong, D.G., Wang, Y.L., Ma, J., Klosterman, S.J., Xiao, S., Tian, C. Deep mRNA sequencing reveals stage-specific transcriptome alterations during microsclerotia development in the smoke tree vascular wilt pathogen, Verticillium dahliae. BMC Genomics 2014, 15(1): 324, doi: 10.1186/1471-2164-15-324.
  7. Fan, S.S.; Huang, Y.J.; Zhang, X.J.; Chen, G.H.; Zhou, J.; Li, X.; Han, M.Z. First report of Botryosphaeria dothidea causing canker on Cotinus coggygria in China. Plant Disease 2019, 103, 2678-2678, doi:10.1094/pdis-04-19-0690-pdn.

Point 13: “Further analysis showed that the species in the genus Lasiodiplodia sampling from China tend to clustered together (Table 1 and Figure 1).”What is your opinion on this?

Response 13: Thank you for your question. The Lasiodiplodia species on different hosts sampling from China tend to clustered together, which may be the result of comprehensive action of fungal adaptive ability, regional climate and human-mediated factors. The identification and characterization of novel fungal taxa and new host records is an indication of the high potential to evolve rapidly. Host switching is often related to fungal adaptive ability [1]. The changing environments and human interference present both challenges and opportunities for fungi, with some capable of switching from endophytic or saprobic lifestyles to pathogenic styles or becoming more aggressive and colonizing new hosts [2]. During the past decade, northern China has become significantly warmer [3]. The increased temperature could attract new fungi to the region. On the other hand, human-mediated factors can also influence the development of a new fungi [4]. Studying the genetic diversity of fungi provides clues to how host switches might have occurred and the genetic basis for new fungus emergence.

  1. Bleuven, C., and Landry, C. R. (2016). Molecular and cellular bases of adaptation to a changing environment in microorganisms. R. Soc. 283:20161458. doi: 10.1098/rspb.2016.1458
  2. Manawasinghe, I. S., Zhang, W., Li, X., Zhao, W., Chethana, K. T., Xu, J., et al. (2018). Novel microsatellite markers reveal multiple origins of Botryosphaeria dothidea causing the Chinese grapevine trunk disease. Ecol. 33, 134–142. doi: 10.1016/j.funeco.2018.02.004
  3. Piao, S., Ciais, P., Huang, Y., Shen, Z., Peng, S., Li, J., et al. (2010). The impacts of climate change on water resources and agriculture in China. Nature 467, 43–51. doi: 10.1038/nature09364
  4. Úrbez-Torres, J. R. (2011). The status of Botryosphaeriaceae species infecting grapevines. Phytopathol. Mediterr. 50, 5–45. doi: 10.14601/Phytopathol_Mediterr-9316 

Round 2

Reviewer 3 Report

This manuscript "Two Novel Lasiodiplodia Species from Blighted Stems of Acer truncatum and Cotinus coggygria in China" was revised properly based on the comments given.